# Deep Learning and Handcrafted Features for Virus Image Classification

**DOI:** 10.3390/jimaging6120143

**Published:** 2020-12-21

**Authors:** Loris Nanni, Eugenio De Luca, Marco Ludovico Facin, Gianluca Maguolo

**Affiliations:** Dipartimento di Ingegneria Dell’informazione, University of Padova, via Gradenigo 6, 35131 Padova, Italy; eugenio.deluca@studenti.unipd.it (E.D.L.); marcoludovico.facin@studenti.unipd.it (M.L.F.); gianluca.maguolo@phd.unipd.it (G.M.)

**Keywords:** virus classification, texture descriptors, deep learning, local binary patterns, ensemble of descriptors

## Abstract

In this work, we present an ensemble of descriptors for the classification of virus images acquired using transmission electron microscopy. We trained multiple support vector machines on different sets of features extracted from the data. We used both handcrafted algorithms and a pretrained deep neural network as feature extractors. The proposed fusion strongly boosts the performance obtained by each stand-alone approach, obtaining state of the art performance.

## 1. Introduction

Recognizing and classifying viruses is fundamental to the medical field for both diagnosis and research. Since this task requires highly qualified medical-staff, there is a growing interest in making this process automatic. Such images can be acquired using electronic microscopy, which are currently not used in clinical practice, and that could be an innovative diagnostic tool.

The main difficulty in classifying viruses is their large number, due to the introduction of the DNA sequencing technique that made the number of classified viruses grow exponentially.

Besides, there are other factors that make the creation of an accurate virus taxonomy very complex: their replication and genetic heritage.

The availability of virus datasets acquired using transmission electron microscopy allowed the computer vision research community to look for methods for the automatic classification. This task has been widely investigated over the last decades, and the approaches to its solution followed the same path of the computer vision research. At first, researchers tried to use methods based on training classifiers on descriptors extracted using handcrafted algorithms [1,2,3,4,5,6]. However, the increasing popularity of deep learning led to the training of end-to-end methods that used convolutional networks [7].

In general, the application of machine learning in the field of virus classification and in medicine in general is still evolving and reaching higher and higher accuracies: as new machine learning techniques becoming increasingly reliable, they can become able to replace highly skilled personnel in these tasks.

In this paper, we create a large and effective ensemble of descriptors for virus classification from transmission electron microscopy, combining both handcrafted features and deep learning.

A large set of handcrafted features is used for training a set of support vector machines (SVM), which are then combined by sum rule; another SVM is trained using the features extracted by the last average pooling layer of DenseNet201 neural networks pre-trained on ImageNet and then tuned on the virus images.

The experimental results confirm the validity of the proposed method, since our proposed ensemble obtains state of the art performance in the tested dataset. The MATLAB code used in this study is available at https://github.com/LorisNanni.

The rest of the paper is organized as follows. In Section 2, we summarize some of the most relevant papers in the field. In Section 3, we describe the different approaches used to feed support vector machine (SVM) classifiers. In Section 4, the experimental results are reported and in Section 5, some conclusions are drawn.

## 2. Related Work

One of the first applications of computer vision aimed at the study of viruses is described in [1], in which ring filters are used for feature extraction on a dataset containing images of four types of viruses. The method proposed in [1] uses a Bayesian classifier with an overall accuracy of 98% in the worst case, although on a simple dataset.

In [2], researchers tried to use the features extracted from local binary patterns (LBP) and local directional patterns (LDP) to train a random forest. They obtained an average error of 13% on the same dataset used in this paper. In [3], a new feature extractor for polyomavirus images is introduced in order to automate its segmentation process.

In 2015, innovative feature descriptors such as multi quinary, edge and bag of features, which we shall explain and test below, were proposed in [4] to improve the performance of the FUSION ensemble described in [5], leading to a higher accuracy. The following year, Wen et al. proposed the combination of multi-principal component analysis (PCA) and multi-completed local binary patterns (LBP) and obtained an overall accuracy of 86.20% on the same dataset used in this study [6].

More recent studies proposed the application of transfer learning using pre-trained convolutional neural networks (SqueezeNet, DenseNet, ResNet, and InceptionV3) [7] obtaining very promising results.

In 2020, in [8], Backes et al. proposed the application of various extractors (such as Gabor wavelets, randomized neural network signature, and others) on the same dataset used in this work with promising results: the combination of all these methods led to an average accuracy of 87.27%.

Still in 2020, Wen et al. applied a method based on the use of filtering images through principal component analysis and Gaussian filters, obtaining an overall accuracy of 88%, which was the state of the art performance when the paper was published.

## 3. The Proposed Method

This section describes the methods tested in this paper and those used in previous works. Every method returns a feature vector that describes the image and a support vector machine classifies it. Finally, we provide a description of the deep learning approach. To avoid misunderstandings on parameter values, the code of each descriptor is available on https://github.com/LorisNanni.

### 3.1. Method Description and SVMs

We use a large number of texture descriptors available in the literature. We then train a SVM on these descriptors and we evaluate their stand-alone performance.

SVMs are classical tools in machine learning. A SVM is a trainable classifier that finds a separating hyperplane among the samples in a high dimensional space by maximizing the distance between data points and the hyperplane. They are used in two label classification problems, however, they can be generalized to *n* class problems by training a set of SVMs, each of which is trained to detect whether a samples belongs to a given class. Each SVM returns a probability that a sample belongs to that specific class. The output of the set of SVMs is the one with the largest probability. We refer the reader to [9] for a more detailed introduction to SVMs.

We evaluate the accuracy reached by the ensemble of those SVMs. We combine the single results of the SVMs using the sum rule, which consists in summing all the output probability vectors of the SVMs and defining the output class of a sample as the one with the highest sum of the scores. Besides, we also train a SVM on the features extracted by a deep learning architecture. We shall now describe these methods.

### 3.2. Texture Descriptors Tested in This Paper

Here, we shall describe the texture descriptors that we used in this paper and the ones used in the papers that we compare with.

#### 3.2.1. Local Binary Pattern (LBP)

LBP operator provides a descriptor for any image using the gray levels of each pixel. In its classic version taken, a pixel is considered the 8 adjacent to him, thus creating a square of 3 × 3 pixels.

Then, for each of these 8 neighbors, their relationship with the central pixel is evaluated: if their grey level is greater than that of the central pixel, they are replaced by a 1, otherwise by a 0.

The resulting binary pattern is then converted to a decimal number. The *LBP* operator is thus evaluated:(1)$$LBP\left(x_{c},y_{c}\right)=\underset{p\in P}{\sum }2^{p}*q\left(i_{p}-i_{c}\right),$$
where *P* is a structure that describes pixels close to the central one, *i_c_* and *i_p_* are respectively the grey levels of the central pixel and its *p*-th neighbor, and *q(z)* is a quantization function defined as:(2)$$q\left(z\right)=\{\begin{array}{l} 1,\;z\geq 0\\ 0,\;otherwise \end{array}.$$

It is worth noticing that the number of neighbors and the radius are parameters that can be modified at will and are not static.

#### 3.2.2. Discrete Local Binary Pattern (DLBP)

Described by Takumi Kobayashi [10], this algorithm is a modified version of local binary pattern.

The idea is to find, for each pixel patch, the best threshold that divides the pixels inside it. This threshold is obtained by minimizing a residual error calculated as such:(3)$$\varepsilon \left(\tau \right)=\frac{1}{N}\left\{\underset{i|I\left(r_{i}\right)\leq \tau }{\sum }{\left(I\left(r_{i}\right)-\mu_{0}\right)}^{2}+\underset{i|I\left(r_{i}\right)>\tau }{\sum }{\left(I\left(r_{i}\right)-\mu_{1}\right)}^{2}\right\}.$$

With:(4)$$\mu_{0}=\frac{1}{N_{0}}\underset{i|I\left(r_{i}\right)\leq \tau }{\sum }I\left(r_{i}\right)\;and\;N_{0}=\underset{i}{\sum }q\left(\tau -I\left(r_{i}\right)\right),$$
(5)$$\mu_{1}=\frac{1}{N_{1}}\underset{i|I\left(r_{i}\right)>\tau }{\sum }I\left(r_{i}\right)\;and\;N_{1}=\underset{i}{\sum }q\left(I\left(r_{i}\right)-\tau \right).$$

The residual error *ε(τ)* is then configured as the intra-class variance *σ^2^_W_*_._ The best threshold is the one that maximizes the variance between classes *σ^2^_B_* of the pixels in the patch calculated as such:(6)$$\sigma_{B}^{2}=\sigma -\sigma_{W}^{2}.$$

Once this threshold is found, the weight of each pixel patch on the final histogram is calculated:(7)$$\omega =\sqrt{\frac{\sigma_{B}^{2}\left(\gamma^{*}\right)}{\sigma^{2}+C}},$$
where *C* is a constant that serves to handle those cases where *σ*^2^ is close to zero and may cause the weight of the vote to fluctuate too much. In Kobayashi’s paper, it is placed at 0.01^2^. The feature is extracted with (radius, neighbors) = (1, 8), (2, 8), (3, 8).

#### 3.2.3. Sorted Consecutive Local Binary Pattern (scLBP)

The sorted consecutive local binary pattern (scLBP) algorithm has been described by Jongbin Ryu et al. during 2015 in [11].

This particular approach tries to solve some inconsistencies in the rotation invariant local binary pattern. In scLBP, from each pixel patch are extracted four components: SCLBP_S, SCLBP_M+, SCLBP_M−, SCLBP_C.

These components are then encoded by counting how much consecutive 0 s and 1 s there are and saving the result in two different array that will be sorted and concatenated.

For example, the pattern {00111010} will be encoded in {3, 1, 0} and {2, 1, 1} and then in {3, 1, 0, 2, 1, 1}. After this step, the histogram is obtained through dictionary learning with kd-tree on the raw feature of each pixel of the image, however, in our study, we used k-means with 255 clusters instead of the kd-tree.

Center pixel is a 16 element cell vector, each cell contains a matrix of centroids for each possible radius (for radius equal to 1 to 4 in steps of 0.2, so 16 elements).

#### 3.2.4. Attractive Repulsive Center Symmetric Local Binary Pattern (ARCSLBP)

ARCSLBP is a LBP variant proposed by El Merabet et al. [12], from a grey scale portion of image, it considers four triplets around the center pixel (Figure 1a) and average local gray level (ALGL), average global grey level (AGGL), and the mean value of the considered neighbors.

As can be seen in Figure 1b, the pattern of single ACSLBP is shorter than standard LBP; in the 8 neighbors case, the LBP method has as a result, a histogram with 2^8^ possible patterns, ACSLBP only 2^7^.

The final ARCSLBP length is 2^8 (2^7 ACSLBP + 2^7 RCSLBP), which is the same of LBP but with improved performance as proved by El Merabet et al. [12].

In this paper, ARCSLBPrn is proposed with the variant of radius and neighbors, like standard LBP, the number of triplets can go up to half the neighbors (for example 8 triplets in 16 neighbors case), also average local gray level (ALGL) and average neighbors value are affected. The final histogram length is 2^(half of the neighbors + 3 pixel means). The feature is extracted with (radius, neighbors) = (1, 8), (2, 8), (3, 8).

#### 3.2.5. Sigma Attractive Repulsive Center Symmetric Local Binary Pattern (sigmaARCSLBP)

Alpaslan et al. [13] merged the effectiveness of ARCSLBP with the Hessian matrix directional derivative information:(8)$$\begin{array}{cc} H\left(I\left(x,y\right)\right)= & [I_{xx}=\frac{d^{2}I\left(x,y\right)}{dx^{2}}I_{xy}=\frac{d^{2}I\left(x,y\right)}{dxdy}I_{yx}=\frac{d^{2}I\left(x,y\right)}{dxdy}I_{yy}\\ & =\frac{d^{2}I\left(x,y\right)}{dy^{2}}], \end{array}$$
(9)$$\begin{array}{c} G_{xx}=\frac{1}{2\pi \sigma^{4}}\left(\frac{x^{2}}{\sigma^{2}}-1\right)e^{\left(-\frac{x^{2}+y^{2}}{2\sigma^{2}}\right)}\\ G_{yy}=\frac{1}{2\pi \sigma^{4}}\left(\frac{y^{2}}{\sigma^{2}}-1\right)e^{\left(-\frac{x^{2}+y^{2}}{2\sigma^{2}}\right)}, \end{array}$$
(10)$$\begin{array}{c} I_{xx}=I*G_{xx}\\ I_{yy}=I*G_{yy}\\ Mag=\sqrt{I_{xx}^{2}+I_{yy}^{2}}. \end{array}$$

The magnitude information of the Hessian matrix with different σ variance values is used in the ACS-LBP method with variable radius and neighbors. The magnitude is useful to identify intense or flat portions of image (Figure 2). The feature is extracted with (radius, neighbors) = (1, 8), (2, 8), (3, 8).

#### 3.2.6. Alpha Local Binary Pattern (alphaLBP)

Proposed by Kaplan et al. [14], the alpha LBP operator calculates the value of each pixel based on an angle value α. The code is the same of standard LBP, although the neighbors considered are on a line as shown in Figure 3a.

This method has similar performance with standard LBP and the same pattern length, but it is helpful to find image micro pattern. Like in LBP from output image (Figure 3b), a histogram of pattern frequency is extracted. In this study, we use an 8 neighbors line.

#### 3.2.7. Heterogeneous Auto-Similarities of Characteristics (HASC)

HASC [15] combines linear relations by covariances (COV) and nonlinear associations with entropy combined with mutual information (EMI) of heterogeneous dense feature maps.

Covariance matrices are used as descriptors because they are low in dimensionality, robust to noise and, the covariance among two features describes the features of the joint PDF.

The entropy (E) of a random variable measures the uncertainty of its value, and the mutual information (MI) of two random variables captures their generic linear and nonlinear dependencies. HASC divides the image in portions, it calculates the EMI matrix and then concatenates the vectorized EMI and COV.

#### 3.2.8. Local Concave Micro Structure Pattern (LCvMSP)

Differently from LBP, LCvMSP (Merabet et al. [16]) calculates the relation between center pixel and neighbors mathematically.

This method uses the median value of the 3 × 3 neighbors and the entire image in both normal e grayscale value, these new two statistical triplets are used in LCvMSP.

Adding these two extra bits to concave binary thresholding function 1023 different pattern result, with improved statistical information (Alpasan et al. [17]).

#### 3.2.9. JET Texton Learning

This method extract six derivative of Gaussian (DtGs) for every pixel forming a six dimensions feature vector (jet vector), then k-means clustering is used to construct a jet texton dictionary (Roy et al. [18] for exhaustive information). The resulted is a histogram of the jet texton. Jet has excellent classification performance in large datasets.

#### 3.2.10. Adaptive Hybrid Patterns (AHP)

Introduced by Zhu et al. [19], it combines a hybrid texture model (HTD) and adaptive quantization algorithm (AQA). HTD is composed of local primitive features and global spatial structure and AQA improves noise robustness.

### 3.3. Texture Descriptors Proposed in the Literature

A selection of descriptors from Santos et al. [4].

#### 3.3.1. Local Ternary Pattern (LTP)

LTP (Tan and Triggs [20]) is a simple LBP variant in which a threshold is set in the comparison, this helps with different light conditions in a uniform area and provides better discrimination power, this allows a certain amount of noise before binarizing between the neighbor and the central pixel (Santos et al. [4]).

For every couple of pixel considered, 3 values are possible (function *s*(*x*)), with 0 if the difference is minor of the threshold *τ*, with a resulting 3^(number of neighbors) histogram length. Santos decodes to divide the LBP + e LBP-parts in two 2^(number of neighbors) histograms and then to concatenate them:(11)$$s\left(x\right)=\{\begin{array}{ll} 1, & x\geq \tau \\ 0, & \tau \leq x<\tau \\ -1, & otherwise \end{array}.$$

#### 3.3.2. Local Phase Quantization (LPQ)

LPQ (proposed by Ojansivu [21]) uses the blur invariance property of the Fourier phase spectrum. It considers a rectangular neighborhood where it computes the 2D short term Fourier transform (STFT) to extract local phase information.

#### 3.3.3. Rotation Invariant Co-Occurrence among Adjacent LBP

RI (Nosaka, Ohkawa, and Fukui [22]) is a version of LBP that considers the spatial co-occurrence of the feature codes.

Each occurrence pair is labeled by the binary code and a spatial vector which connects the two center pixels. Radius and neighbors are fixed as Santos et al. [4].

#### 3.3.4. Local Binary Pattern Histogram Fourier

LHF (Ahonen et al. [23]) uses the rotation of the LBP neighbors of an angle, which is a multiple of 360/P where P is the number of neighbors.

The algorithm uses the fact that if the image is rotated, then the neighbors will be rotated by the same angle. Radius and neighbors are variable (Santos et al. [4]).

#### 3.3.5. Dense LBP (DLBP)

DLBP (Ylioinas et al. [24]) use the same neighbors of LBP, plus the corners between centered on the corner between center pixels.

It results in longer histogram to avoid noise. Radius and neighbors are variable (Santos et al. [4]).

#### 3.3.6. Multi Quinary Coding (MQC)

This method extends LTP function *s*(*x*) with 2 threshold *τ*, *θ* there are 5 possible outputs:(12)$$f\left(x\right)=\{\begin{array}{ll} 2, & x\geq \tau \\ 1, & \theta \leq x<\tau \\ 0, & 0\leq x<\theta \\ -1, & -\theta \leq x<0\\ -2, & otherwise \end{array}.$$

Like LTP, the labels are split into 4 binary patterns to reduce verbosity (Paci et al. [25]).

#### 3.3.7. Edge (ED)

The idea behind the ED method is to focus on the important portions of an image and apply a LBP like approach. Those salient regions are the edge and non-edge and we find them where the gradient function is higher. ED is used to create an ensemble of LBP like methods, for details read Santos et al. [4].

#### 3.3.8. Difference of Gaussian (DoG)

DoG indicates the convolution of the original image with a 2D dog filter obtained by the subtraction of two images blurred by different Gaussian kernel with different variance, the result is similar to a band pass filter, for details read Santos et al. [4].

#### 3.3.9. Bag of Feature (BoF)

BoF divides the images into regions and then it extracts different features from them, in order to build a visual vocabulary. From a new image, feature vectors are extracted and assigned to the nearest matching terms from the vocabulary, for details read Santos et al. [4].

### 3.4. Deep Learning Approach

Deep learning and neural networks revolutionized the field of machine learning. They can be implemented using several different algorithms, all of which consist of a cascade of many processing layers organized in a hierarchical structure.

Each of these layers adds a level of abstraction to the overall representation. In the image interpretation task, layers close to the input deal with low-level features like edges and texture.

These low-level features can be combined together to build a more complex representation. Layer by layer, complexity increases.

The approach to deep learning considered in this paper is based on convolutional neural networks (CNNs) [26].

In this paper, we use a version of DenseNet201 [27] pretrained on ImageNet. DenseNet is an evolution of ResNet which includes dense connections among layers: each layer is connected to each following layer in a feed-forward fashion. Therefore, the number of connections increases from the number of layers L to L × (L + 1)/2. DenseNet improves the performance of previous models at the cost of an augmented computation requirement. It accepts images of 224 × 224 pixels.

We used the following hyperparameters for training: 50 training epochs, mini-batch size of 30 observations, learning rate of 0.001.

As data augmentation protocol, we independently reflect the images in both the left-right and the top-bottom directions with 50% probability. We also linearly scale the image along both axes by two random numbers in [1,2].

We use a trained Densenet as feature extractor and we take the last average pooling layer as the output of our network. We then train a SVM on these extracted features.

## 4. Results

As already mentioned in the introduction, the dataset used can be found at the following link: http://www.cb.uu.se/_Gustaf/virustexture/ and is described by Kylberg et al. [2]. It contains 1500 transmission electron microscopy (TEM) images of size 41 × 41 of viruses belonging to 15 different species (specifically: Adenovirus, Astrovirus, CCHF, Cowpox, Dengue, Ebola, Influenza, Lassa, Marburg, Norovirus, Orf, Papilloma, Rift Valley, Rotavirus, Westnile). We see some examples in Figure 3.

In Table 1, we report the performance of the texture descriptors here tested. Clearly, the performance is very different considering different descriptors.

We now report the results obtained by Densenet and by the fusion of multiple methods. In order to combine the results of multiple classifiers, we use the sum rule, i.e., we sum the output scores of all the classifiers in the ensemble and then the class selected by the ensemble is the one with the highest sum of the scores.

In Table 2, we report the accuracies of following ensemble methods:-NewSet consists of the sum rule among the methods reported in Table 1. It is interesting to note that the fusion strongly outperforms the stand-alone approaches.-OLD is the previous set proposed in Santos et al. [4].-HandC is the fusion by sum rule among the handcrafted methods of NewSet and OLD. This ensemble does not boost the performance of NewSet significantly.-DeepL is the SVM trained using the last average pooling layer as input. Notice that using DenseNet201 as classifier, a lower 78.93% accuracy is obtained.-HandC+DeepL is the sum rule between HandC and DeepL. Before the fusion, the scores of HandC and DeepL are normalized to mean 0 and standard deviation 1. This is because the number of classifiers in HandC and DeepL are different.

Finally, in Table 3, we compare our approach with other methods already reported in the literature using the same dataset and the same testing protocol.

In [2], the results in RDPF e RDPF + LBPF are obtained on the fixed scale version of our dataset that is not available online so a comparison is not possible. In the fixed version image, 1 pixel corresponds to 1 nanometer, the viruses in this study have a diameter from 25 to 270 nm, object scale image is always resized to 41 × 41 pixels.

Our ensemble obtains performance comparable with the best result already published. We want to remark that our Densenet does not reach the accuracy reported in [7].

## 5. Conclusions

In this paper, we proposed an ensemble of handcrafted descriptors and deep learning features for virus classification acquired using transmission electron microscopy.

For each descriptor, a different SVM is trained. We combined several sets of SVMs by sum rule. Our largest ensemble obtains state of the art performance on a very competitive dataset. This shows that combining handcrafted descriptors and deep learning features allows to boost the performance that can be obtained using only handcrafted descriptors or deep learning.

The MATLAB code of the proposed ensemble is available at https://github.com/LorisNanni.

## Figures and Tables

**Figure 1 jimaging-06-00143-f001:**
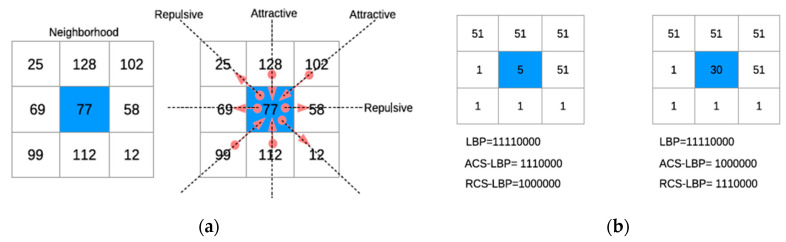
El Merabet et al. [12] representation of Attractive Repulsive Center Symmetric Local Binary Pattern: (**a**) Example of possible triplets center symmetric; (**b**) Example of various methods patterns.

**Figure 2 jimaging-06-00143-f002:**
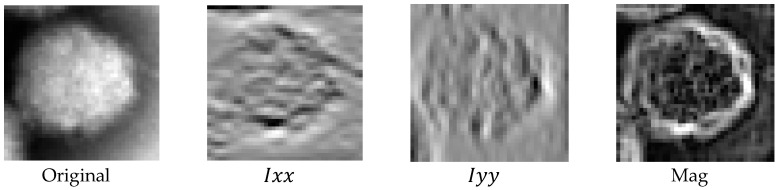
Example of magnitude with Gaussian standard deviation σ = 1 (Lassa Virus).

**Figure 3 jimaging-06-00143-f003:**
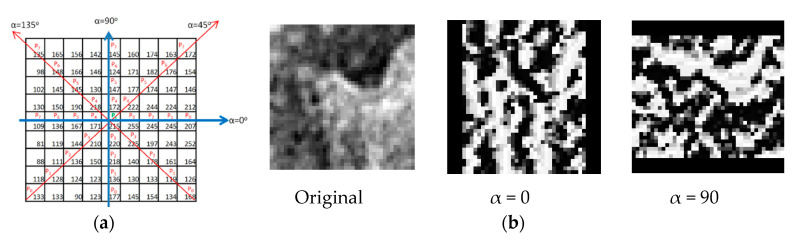
Kaplan et al. [14] proposed method. (**a**) Angle based neighbors. (**b**) Example of application on a TEM image of the influenza virus with various angles.

**Table 1 jimaging-06-00143-t001:** Accuracy of the texture descriptors.

JET	scLBP	AHP	HASC	Gradient + ARCSLBP	ARCSLBP	AlphaLBP	SigmaARCSLBP	DLBP	LCvMSP
58.93	69.47	76.60	68.40	61.00	79.93	64.13	75.40	70.33	64.67

**Table 2 jimaging-06-00143-t002:** Performance of the ensemble.

NewSet	OLD	HandC	DeepL	HandC + DeepL
85.40	85.67	86.13	86.40	89.47

**Table 3 jimaging-06-00143-t003:** Comparison with the literature on the object scale dataset.

Here 2020	DenseNet [7] 2020	PCA [28] 2018	Fusion [8] 2020	LBPF [2] 2011 Fixed Scale	RDPF [2] 2011	RDPF + LBPF [2] 2011 Fixed + Object Scale	MPMC [6] 2016	NewF [4] 2015
89.47	89.00	88.00	87.27	79.00	78.00	87.00	86.20	85.70

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
