# Peer review of "Deep Learning and Handcrafted Features for Virus Image Classification"

_2313-433X, 2020, doi:10.3390/jimaging6120143_

Round 1
Reviewer 1 Report
In this paper, we have proposed an ensemble of handcrafted descriptors and deep learning features for the classification of transmission electron microscopy images of viruses. For each descriptor a different SVM is trained, the set of SVMs is combined by sum rule, the proposed ensemble obtains state of the art performance in largely used dataset: clearly to combine handcrafted descriptors and deep learning features permits to boost the performance that can be obtained using only handcrafted descriptors or deep learning. I have the following concern regarding the work:
How does handcrafted features depend on the quality of the image? Many handcrafted features are depend on the image resolution, noise and number of voxels. How these features are reliable in clinical context ?
In introduction: "The method proposed in [1] uses a Bayesian classifier with an overall accuracy of 98% in the worst case. " has been written. If they already got that accuracy then what is your contribution or what problems you want to solve is not clear in the paper.
Support Vector Machine: explained here is just conventional method. I am not clear what modification or update has been done on the Kernel of SVM ?
The equation in Local Ternary Pattern (LTP) is not visible. Authors must give the equation number in each equation to clearly refer the equations.
I did not find any novelty of the paper other than performance comparison of existing methods.
English is not very clear among the different paragraph of the paper. Authors should improve from starting to end of the paper.
Author Response
In this paper, we have proposed an ensemble of handcrafted descriptors and deep learning features for the classification of transmission electron microscopy images of viruses. For each descriptor a different SVM is trained, the set of SVMs is combined by sum rule, the proposed ensemble obtains state of the art performance in largely used dataset: clearly to combine handcrafted descriptors and deep learning features permits to boost the performance that can be obtained using only handcrafted descriptors or deep learning. I have the following concern regarding the work:
How does handcrafted features depend on the quality of the image? Many handcrafted features are depend on the image resolution, noise and number of voxels. How these features are reliable in clinical context ?
We use a very competitive dataset, whose resolution is high enough for an accurate classification. Handcrafted features depend on the resolution of the images, hence it is not trivial to say in advance which ones will be the most effective. Hence, our ensemble method increases the possibility the some of the handcrafted features are well suited for the resolution of the dataset. Experimental results show that, in our setting, they work very well. As far as it concerns the problem of testing on images with different resolution, this is a domain shift problem that is challenging for all classifiers, and even for deep networks. In general, we can expect our models to perform well on high resolution images, since they are resized to 224x224 in order to be fed into Densenet, and this leads to a resolution which is similar to the images used in the training set. Low resolution images might lead to classification problems, however modern instruments acquire images at a resolution which is high enough for our purposes.
In introduction: "The method proposed in [1] uses a Bayesian classifier with an overall accuracy of 98% in the worst case. " has been written. If they already got that accuracy then what is your contribution or what problems you want to solve is not clear in the paper.
The dataset used in that paper is very different from the one that we used and it only has four classes. Nowadays, more challenging datasets are used. We now explain that it is a simple dataset.
Support Vector Machine: explained here is just conventional method. I am not clear what modification or update has been done on the Kernel of SVM ?
We removed the long description of SVMs and we now introduce them in Section 3.1. We refer the reader to the literature for the details, we only explain of we generalize them to n classes.
The equation in Local Ternary Pattern (LTP) is not visible. Authors must give the equation number in each equation to clearly refer the equations.
We fixed the problem and numbered the equation
I did not find any novelty of the paper other than performance comparison of existing methods.
In this paper we show that our ensemble of handcrafted features and deep learning models outperform every stand-alone approach and sets a new state of the art on a competitive dataset and at an important task like virus classification. Besides, we tested several algorithms as baselines and we replicate the experiments made in the literature, which is important in scientific research. Last but not least, we share all our code for encourage to reproduce our results.
English is not very clear among the different paragraph of the paper. Authors should improve from starting to end of the paper.
The suggestion is fair, we carefully read the paper and drastically changed the sections that were not clear or that did not contain an appropriate language for a scientific publication. We hope that now it satisfies the required standards.
Reviewer 2 Report
Abstract: Please consider editing the language of the abstract. It is quite chaotic. An example: “The set of handcrafted is mainly based on Local Binary Pattern variants, for each descriptor a different Support Vector Machine is trained, then the set of classifiers is combined by sum rule. The deep learning approach is a densenet201 pretrained on ImageNet and then tuned in the virus dataset, the net is used as features extractor for feeding another Support Vector Machine, in particular the last average pooling layer is used as feature extractor.” Please take into consideration that the audience of the Journal of Imaging is not only constituted by specialist in computer science but also physicians and other clinical bodies.
Page 1, Lines 24-26: Please consider adding that not only highly qualified medical-staff is needed but also imaging modalities. Such ones that are still not being used currently in daily clinical practise, such as electron microscopes.
Page 1, Lines 27-33: This paragraph can be reduced significantly by just mentioning that there is few research done in the topic. As in the next paragraphs the authors mention previous research in this topic they can be fused into a single, simpler and more linear, paragraph.
Page 2, Line 56: Please be careful using the term AI so freely. ML/DL does not encompass the whole AI.
Pages 2-4 Lines 76-160: May I ask what the use is of having such in-depth analysis of viruses additive to this study? I would suggest that the authors completely discard this part of the manuscript.
Please consider reducing the size of the explanation of SVM, it is not necessary for a technical reader and a clinical reader is not interested in such in-depth analysis.
It seems that there are tables missing. Please include them for revision.
Overall comments:
- Please consider passing the current manuscript draft through a rigorous editorial (or even developmental editing) check-up. The text is both incomprehensive in several passages and also confusing. There is also a high number of colloquial expressions that should be rewritten. For example: “Otherwise you can do what is called Kernel Trick,…”
- I have been following Professor Nanni’s works throughout the years. Conciseness was a hallmark of his previous works. Clearly, this manuscript was not written by him, and it is not to his high level of expertise. Most likely, it has been drafted by one of his students. Actually, based on quality, quantity of images, the mixture of introduction/methods/materials, and the presence of the deep explanation of “what viruses are”, this manuscript seems to be an abbreviation or abstract of a M.Sc. thesis.
Author Response
Abstract: Please consider editing the language of the abstract. It is quite chaotic. An example: “The set of handcrafted is mainly based on Local Binary Pattern variants, for each descriptor a different Support Vector Machine is trained, then the set of classifiers is combined by sum rule. The deep learning approach is a densenet201 pretrained on ImageNet and then tuned in the virus dataset, the net is used as features extractor for feeding another Support Vector Machine, in particular the last average pooling layer is used as feature extractor.” Please take into consideration that the audience of the Journal of Imaging is not only constituted by specialist in computer science but also physicians and other clinical bodies.
Thanks. The abstract has been rewritten almost completely, taking the suggestion into account.
Page 1, Lines 24-26: Please consider adding that not only highly qualified medical-staff is needed but also imaging modalities. Such ones that are still not being used currently in daily clinical practise, such as electron microscopes.
Thanks. We added the line: “Such images can be acquired using electronic microscopy, which are currently not used in clinical practice and that could be an innovative diagnostic tool.”
Page 1, Lines 27-33: This paragraph can be reduced significantly by just mentioning that there is few research done in the topic. As in the next paragraphs the authors mention previous research in this topic they can be fused into a single, simpler and more linear, paragraph.
It is true, we created a Section for the related work and moved those lines to the new section.
Page 2, Line 56: Please be careful using the term AI so freely. ML/DL does not encompass the whole AI.
We now use the term machine learning, which is more precise.
Pages 2-4 Lines 76-160: May I ask what the use is of having such in-depth analysis of viruses additive to this study? I would suggest that the authors completely discard this part of the manuscript.
This is true. We deleted those parts, but we moved a small part of them in the introduction, just to explain the difficulties of this particular task.
Please consider reducing the size of the explanation of SVM, it is not necessary for a technical reader and a clinical reader is not interested in such in-depth analysis.
We now explain them in a short paragraph in Section 3.1. We refer the reader to the literature for a more detailed introduction to SVMs.
It seems that there are tables missing. Please include them for revision.
Now everything should be included in the paper.
Overall comments:
- Please consider passing the current manuscript draft through a rigorous editorial (or even developmental editing) check-up. The text is both incomprehensive in several passages and also confusing. There is also a high number of colloquial expressions that should be rewritten. For example: “Otherwise you can do what is called Kernel Trick,…”
We carefully read the paper and made several changes. We believe that the language used is now adequate.
- I have been following Professor Nanni’s works throughout the years. Conciseness was a hallmark of his previous works. Clearly, this manuscript was not written by him, and it is not to his high level of expertise. Most likely, it has been drafted by one of his students. Actually, based on quality, quantity of images, the mixture of introduction/methods/materials, and the presence of the deep explanation of “what viruses are”, this manuscript seems to be an abbreviation or abstract of a M.Sc. thesis.
As we said, we made several changes to the paper. We only decided to keep the description of the handcrafted methods, since technical readers are likely not to know every method that we used. However, those parts might be skipped by those readers that are interested in other aspects of the paper. In the rest of the paper, we now go straight to the point.
Round 2
Reviewer 1 Report
Authors present an ensemble of descriptors for the classification of transmission electron microscopy images of viruses and proposed to combine handcrafted and deep learning approaches for virus image classification. The set of handcrafted is mainly based on Local Binary Pattern variants, for each descriptor a different Support Vector Machine is trained, then the set of classifiers is combined by sum rule. The deep learning approach is a densenet201 pretrained on ImageNet and then tuned in the virus dataset, the net is used as features extractor for feeding another Support Vector Machine, in particular the last average pooling layer is used as feature extractor. Now the paper is in good shape. Authors have addressed all my previous comments carefully.
Reviewer 2 Report
Lines 24-29: „Such images … and genetic heritage.”- There is no previous mention of any kind of images. Please correct.
- “… there was …” means that no longer is of interest, which is false. Please change to “… there is a growing interest …”
- Regarding “The main difficulty … is their large numbers …” it is indeed one of the main difficulties. But, as you propose here an image-based method, to my knowledge, such "difficulty" is no longer applied.
Even if the authors improved considerably the language of the manuscript, I would still suggest improving further. There are several sentences that sound colloquial or are hard to understand.